# INST-IT: Boosting Instance Understanding via Explicit Visual Prompt Instruction Tuning

Wujian Peng[1,2*]   Lingchen Meng[1*]   Yitong Chen[1,2]   Yiweng Xie[1]   Yang Liu[1]
Tao Gui[1,2]   Hang Xu[3]   Xipeng Qiu[1,2]   Zuxuan Wu[1,2†]   Yu-Gang Jiang[1]

[1]Institute of Trustworthy Embodied AI, Fudan University
[2]Shanghai Innovation Institute  [3]Huawei Noah's Ark Lab

https://inst-it.github.io

## Abstract

Large Multimodal Models (LMMs) have made significant breakthroughs with the advancement of instruction tuning. However, while existing models can understand images and videos at a holistic level, they still struggle with instance-level understanding that requires a more fine-grained comprehension and alignment. Instance-level understanding is crucial for LMMs, as it focuses on the specific elements that we are most interested in. Excitingly, existing works find that the state-of-the-art LMMs exhibit strong instance understanding capabilities when provided with explicit visual cues. Motivated by this, we proposed INST-IT, a solution to enhance LMMs in **Inst**ance understanding via explicit visual prompt **I**nstruction **T**uning for instance guidance. INST-IT consists of a benchmark to diagnose multimodal instance-level understanding, a large-scale instruction-tuning dataset, and a continuous instruction-tuning training paradigm to effectively enhance spatial-temporal instance understanding capabilities of existing LMMs. Experimental results show that, enhanced by INST-IT, our models not only achieve outstanding performance on INST-IT Bench and other instance understanding benchmarks, but also demonstrate significant improvements across various generic image and video understanding benchmarks. This highlights that our method not only boosts instance-level understanding but also strengthens the overall capabilities of generic image and video comprehension.

## 1   Introduction

Recently, Large Multimodal Models (LMMs) have seen remarkable advancements. A key breakthrough is visual instruction tuning [45, 17], enabling models to follow any type of user instructions. This paves the way to building general-purpose multimodal assistants capable of handling a wide range of real-world tasks [32]. Inspired by this initial work, numerous follow-up studies have emerged in both image-language [43, 60, 106, 9, 13] and video-language [51, 22, 99, 84, 82] modeling. However, although they can understand images or videos at a holistic level, they still struggle to comprehend instance-specific content that the users are most interested in, as illustrated in Fig. 1 (a).

Instance-level understanding involves comprehending the attributes of individual instances within an image or video, as well as the relationships and interactions between them. This requires models to exhibit nuanced comprehension and fine-grained alignment. Instance understanding has been a long-standing pursuit of the community with extensive efforts devoted to object detection [73, 71, 23, 58], instance segmentation [74, 29, 59], and object tracking [19, 76]. This capability is essential for

---

* Equal contributions; † Corresponding authors.

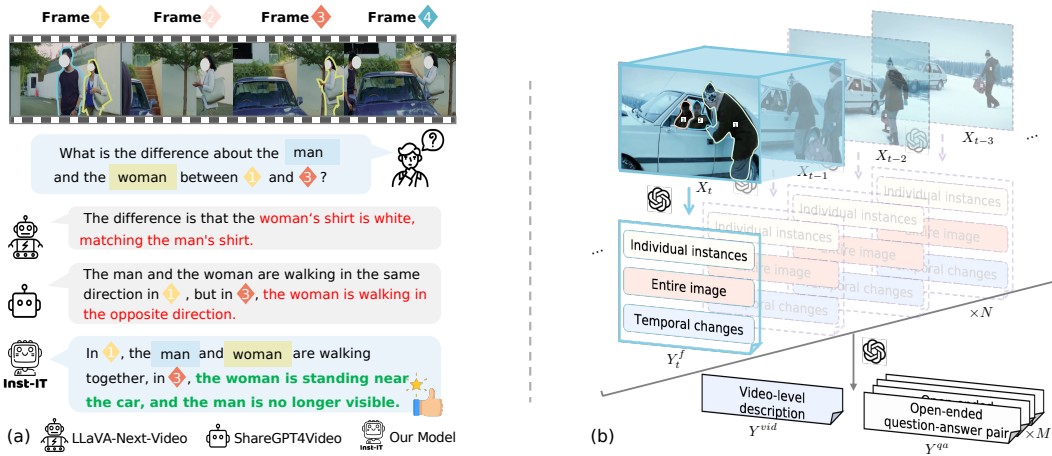

Figure 1: (a) **LMMs struggle with instance understanding**, failing to capture the nuanced details of instances specified in user queries. (b) **Our instance-centric data annotation pipeline**, providing multi-level annotations for individual instances in images and videos.

real-world applications, where users pay more attention to the instances that they are interested in. In the era of LMMs, although there have been some attempts in exploring multimodal instance understanding [24, 5, 101, 105, 102], they are primarily limited in the image domain, leaving the videos under-explored. Compared to images, understanding instances in videos is considerably more challenging, as it requires not only capturing their spatial information but also temporal dynamics. Driven by this, *we aim to advance the multimodal instance understanding in both images and videos.* To this end, we focus on three aspects: instruction-tuning dataset, evaluation benchmark, and training recipe.

Existing multimodal benchmarks and datasets primarily provide coarse-grained knowledge for images and videos, lacking fine-grained annotations for individual instances. To address this, we introduce an automated pipeline to generate detailed instance-specific annotations. As illustrated in Fig. 1 (b), we leverage GPT-4o [61] to produce multi-level annotations, including instance-level descriptions, image-level captions, temporal dynamics, video-level summaries, and open-ended question-answer pairs. To fully unleash the capability of GPT-4o for more accurate annotations, we systematically design task prompts and employ set-of-marks visual prompts [88] to highlight instances in the visual inputs. Powered by this pipeline, we construct INST-IT Dataset, an instance-grounded multimodal dataset comprising 51k images and 21k videos, 207k image-level captions, 135k temporal dynamics descriptions, 21k video-level captions, and 335k open-ended question-answer pairs. Furthermore, we carefully design the INST-IT Bench to diagnose the instance-level understanding capabilities of LMMs, and perform rigorous manual verification and refinement to ensure its data quality.

Building on INST-IT Dataset, we propose a continuous instruction tuning recipe that effectively integrates our instance understanding datasets with general instruction-tuning data. We augment images and videos with visual prompts, and convert the fine-grained annotations from INST-IT Dataset into instruction tuning format, emphasizing the model's spatiotemporal understanding of individual instances. Experimental results show that our enhanced models achieve strong instance understanding performance not only on INST-IT Bench, but also demonstrate consistent improvements on other instance understanding benchmarks e.g. RefCOCOg [53] and ViP-Bench [5]. We also investigate the models' general comprehension capabilities on widely used generic benchmarks. The results reveal significant improvements over the baseline, achieving 4.4% and 13.5% gains on AI2D [28] and ChartQA [54] image benchmarks, as well as 7.8% and 11.8% improvements on Egoschema [52] and NExT-QA [85] video benchmarks, respectively. This highlights the effectiveness of INST-IT in boosting instance understanding while strengthening general comprehension in both images and videos. Our contributions are three-fold:

1. We construct the INST-IT Dataset, the first instance-grounded instruction-tuning dataset that includes both images and videos, featuring explicit instance-level visual prompts and fine-grained annotations grounded on individual instances.

2. We introduce the INST-IT Bench, a human-verified benchmark specifically designed to evaluate the instance-level understanding capabilities of LMMs on both images and videos.

3. We propose a continuous instruction tuning recipe, which leverages our instance-level dataset alongside general data, effectively enhancing models in instance understanding while consistently improving general comprehension in both images and videos.

## 2 INST-IT

To address the scarcity of instance-grounded data, we propose an automated pipeline to generate detailed annotations for both images and videos, with a particular emphasis on ***instances of interest*** (Sec. 2.1). Based on this, we build a large-scale instance-grounded multimodal dataset (Sec. 2.2), and carefully design an instance-centric evaluation benchmark (Sec. 2.3). Furthermore, we propose a continuous instruction-tuning recipe (Sec. 2.4) to enhance LMMs in instance understanding.

### 2.1 Instance-centric annotation pipeline

**Overview.** We propose an automated pipeline to generate annotations grounded on individual instances. As in Fig. 1 (b), we annotate each frame sequentially, aggregate frame-level annotations into a comprehensive video-level description, and generate open-ended question-answer pairs.

**Visual prompting.** Directly processing the raw visual inputs suffers from hallucinations and distraction. To mitigate this issue, we augment the images and videos with visual prompts to highlight the instances. Specifically, we use set-of-marks (SoMs) visual prompt [88], which overlays a numerical ID on each instance. We find this method highly effective in guiding GPT-4o to provide annotations focused on individual instances. For more details, please refer to Sec. A.1.

**Frame-level annotation.** We annotate video frames sequentially. At timestamp $t$, we provide GPT-4o with the current frame $X_t$, the previous frame $X_{t-1}$, and a tailored task prompt $P^f$. We then obtain a frame-level annotation $Y_t^f = (y_t^{ins}, y_t^{img}, y_t^{dif})$ encompassing three aspects, where $y_t^{ins}$ represents the captions for individual instances, $y_t^{img}$ is a caption for the entire image, and $y_t^{dif}$ describes the temporal differences from the previous frame:

$$Y_t^f = GPT(P^f, X_t, X_{t-1}). \tag{1}$$

**Video-level summary.** After obtaining annotations for each frame, we aggregate them into a caption for the entire video $Y^{vid}$, capturing detailed spatiotemporal information of individual instances:

$$Y^{vid} = GPT(P^{vid}, [Y_1^f, Y_2^f, \cdots, Y_N^f]), \tag{2}$$

where $P^{vid}$ is the task prompt designed for video-level summary and $N$ is the total number of frames.

**Open-ended question-answer pairs.** We also prompt GPT-4o with the task prompt $P^{qa}$ to create $M$ open-ended QA pairs $Y^{qa} = \{(q_i, a_i)\}_{i=1}^{M}$ focusing on instance understanding:

$$Y^{qa} = GPT(P^{qa}, [Y_1^f, Y_2^f, \cdots, Y_N^f]). \tag{3}$$

Following these steps, each video is enriched with multi-granularity annotations that incorporate instance-specific information. As illustrated in Fig. 2, these annotations include the following aspects:

- $N$ **frame-level annotations**, each contains detailed descriptions of individual instances, the entire image, and the temporal dynamics between adjacent frames.
- **A comprehensive description** covering the entire video.
- $M$ **open-ended question-answer pairs** that focused on individual instances or their relationships.

Additional information about the design of each task prompt is provided in Sec. A.2.

### 2.2 INST-IT Dataset

Instruction tuning plays a crucial role in multimodal training; however, the lack of instance-level datasets hinders the advancement of instance understanding. Using the data annotation pipeline

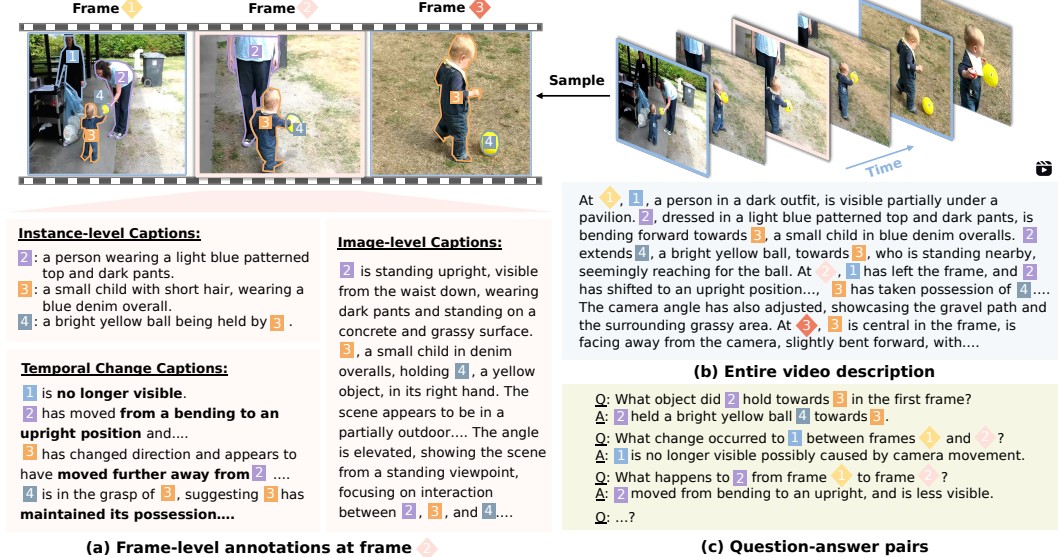

**Figure 2: Visualization of data structure in INST-IT Dataset.** For each video, we provide (a) $N$ frame-level annotations, (b) a video-level description, and (c) $M$ open-ended question-answer pairs. A complete example data can be found in Sec. C.3.

described in Sec. 2.1, we create a large-scale instruction-tuning dataset, the INST-IT Dataset. To the best of our knowledge, this is the first instruction fine-tuning dataset that provides multi-level fine-grained annotations centric on individual instances in both images and videos.

**Data sources.** We utilize five video instance segmentation datasets (BRUST [3, 18], UVO [83], OVIS [65], LVVIS [79] and YoutubeVIS-2021 [89]) and two object tracking datasets (BenSMOT [38], VidOR [75]) as our video sources, as they provide annotations of instance locations, which is useful in SoM visual prompting [88]. For the image source, we select the SA-1B [29] dataset due to its diversity and abundance of instance objects. In total, we collect 51k images and 21k videos. More details can be found in Sec. C.1.

**Statistics.** On average, each video includes one video-level annotation, 7.3 frame-level annotations, and 15.6 open-ended QA pairs. Images are regarded as single-frame videos without temporal changes. In total, INST-IT Dataset includes 21k videos and 51k images, alongside 21k video-level captions, 207k frame-level captions, 836k instance-level captions, 135k temporal descriptions, and 335k open-ended QA pairs. More statistical analyses are provided in Sec. C.2.

**Data quality.** We employ three strategies to ensure the data quality: (1) High-quality visual prompts, we use manually annotated labels in segmentation and tracking tasks as SoMs to reduce noise. (2) Specialized prompt design, we introduce multi-level prompt engineering at the instance, image, two-frame, and video levels to mitigate long-term inconsistencies. (3) Diversity filtering, we filter out samples with few instances to enhance diversity and domain coverage. We randomly select 500 data samples and invite 3 volunteers to independently rate each sample with a score ranging from 1 to 5 (higher is better). The mean$_{\pm\text{std}}$ of scores and average time spent per sample are in Tab. 10. The average score is $4.49_{\pm 0.05}$, indicating the satisfactory quality of our data. We use the maximum score difference (max$_{\text{diff}}$) among volunteers to assess rating consistency. 49.8% of samples have max$_{\text{diff}}$=0, and 78.6% max$_{\text{diff}} \leq 1$, showing high agreements on the ratings of different volunteers.

**Comparison with existing instruction tuning datasets.** Tab. 1 (left) compares INST-IT Dataset with other datasets. Prior video datasets, e.g. ShareGPT4Video [10] and LLaVA-Video [104], focus on holistic understanding without instance-level annotations. While VIP-LLaVA [5] offers instance annotations for images, it does not include any video data. In contrast, INST-IT Dataset encompasses both images and videos with multi-level, fine-grained annotations grounded on individual instances.

Table 1: **Comparison of INST-IT with existing datasets and instance understanding benchmarks.**
**Left:** Instruction tuning datasets. **Right:** Instance understanding benchmarks. IMG and VID indicate
whether the data contains images or videos, respectively. INST denotes the availability of instance-
level annotations. OE and MC indicate open-ended and multiple-choice QA.

| | IMG | VID | INST | | IMG | VID | TASK |
|---|---|---|---|---|---|---|---|
| ShareGPT4Video [10] | | ✓ | | RefCOCO [27] | ✓ | | caption |
| LLaVA-Video [104] | | ✓ | | RefCOCOg [53] | ✓ | | caption |
| ViP-LLaVA-Data [5] | ✓ | | ✓ | ViP-Bench [5] | ✓ | | OE |
| INST-IT Dataset | ✓ | ✓ | ✓ | INST-IT Bench | ✓ | ✓ | OE&MC |

## 2.3 INST-IT Bench

Existing benchmarks primarily focus on global understanding, failing to provide more in-depth
insights into the instance-level comprehension. We present the INST-IT Bench, specifically designed
to diagnose multimodal instance-level understanding in both images and videos.

**Construction process.** To prevent data leakage, we use videos from the test split, ensuring no overlap
with INST-IT Dataset. We apply the pipeline in Sec. 2.1 to generate 20 open-ended QA pairs for each
image and video. Then, we manually review these QA pairs to ensure their accuracy, diversity, and
difficulty. Overly simple questions are removed to ensure the remaining ones are instances-centric.
We also refine the questions and answers, making necessary rephrasing to ensure correctness. After
this rigorous checking process, each sample retains an average of 3.7 carefully polished QA pairs. In
addition, we generate three hard negative for each question to construct a multiple-choice QA data
with four options. More details are provided in Sec. B.1.

**Statistics.** INST-IT Bench comprises 1,000 QA pairs for 338 images and 1,000 QA pairs for 206
videos. Each QA pair supports two evaluation formats, *i.e.* open-ended and multiple-choice.

**Metrics.** For open-ended QAs, we leverage GPT-4o to evaluate the response from a model based on
its similarity to the ground-truth answer. For multiple-choice QAs, we calculate the average accuracy
across all samples. More details about the metric calculations can be found in Sec. B.2.

**Comparison with existing instance understanding benchmarks.** Tab. 1 (right) highlights the
main differences between INST-IT Bench and existing instance understanding benchmarks such as
RefCOCO [27], RefCOCOg [53] and ViP-Bench [5]: (1) its inclusion of evaluation data for both
images and videos, pioneered the evaluation in video LMMs; and (2) it supports both open-ended
and multiple-choice formats, enabling comprehensive evaluation.

## 2.4 Instruction tuning with explicit visual prompt

**Architecture.** We adopt the widely-used LLaVA-NeXT [44] architecture to evaluate the effectiveness
of our INST-IT. We train our model under an image-video joint training pipeline, where we mix our
INST-IT Dataset with the open-source LLaVA-NeXT-DATA [48]. For single-image samples, we
follow the original AnyRes paradigm [44] to split and encode sub-images according to the aspect
ratio. For video and multi-image data, we batch the samples together, encode them, and flatten them
into a sequence. Additionally, we apply $2 \times 2$ spatial pooling to reduce the number of visual tokens
in the video inputs. More details are in Sec. 3.1.

**Converting INST-IT Dataset into instruction tuning format.** INST-IT Dataset provides annotations
at multiple levels of granularity. For the instance- and image-level captions in Fig. 2(a), we use a
single frame as input and structure the task as a two-turn dialogue: the model is first prompted to
describe all individual instances, followed by a holistic description of the entire scene. To capture
temporal dynamics, we use temporal captions from Fig. 2(a), asking the model to describe the
differences between two consecutive frames. The video-level description in Fig. 2(b) is treated as
a captioning task, where the model is instructed to generate a summary based on all video frames.
For the open-ended QA pairs in Fig. 2(c), we organize them into a multi-turn conversation, with the
model answering one question per turn. In total, we construct 243k instruction tuning samples in the
form of single-turn and multi-turn dialogues. All images and video frames are augmented with SoM
visual prompts to explicitly provide instance-level guidance.

Table 2: **Results on INST-IT Bench.** We conduct evaluations on INST-IT Bench, including state-of-the-art open-source image models, video models, and cutting-edge proprietary models. #IT indicates the number of training samples used during the instruction-tuning stage. N/A indicates that the number is unknown. OE and MC represent open-ended and multiple-choice evaluations, respectively.

| Model | LLM | #IT | Image | | Video | |
| --- | --- | --- | --- | --- | --- | --- |
| | | | OE Q&A | MC Q&A | OE Q&A | MC Q&A |
| Random Guess | - | N/A | - | 25.0 | - | 25.0 |
| GPT-4o [61] | - | N/A | 74.1 | 84.8 | 65.5 | 81.0 |
| Gemini-1.5-flash [72] | - | N/A | 65.3 | 79.5 | 57.9 | 75.8 |
| *Open-source image models* | | | | | | |
| LLaVA-1.5 [43] | Vicuna-7B | 665K | 41.6 | 32.1 | - | - |
| ViP-LLaVA [5] | Vicuna-7B | ∼1.2M | 42.1 | 29.2 | - | - |
| SoM-LLaVA [86] | Vicuna-7B | 695K | 45.1 | 40.0 | - | - |
| LLaVA-NeXT [44] | Vicuna-7B | 765K | 46.0 | 42.4 | - | - |
| *Open-source video models* | | | | | | |
| LLaVA-NeXT-Video [103] | Vicuna-7B | 860K | 46.5 | 39.5 | 25.8 | 24.8 |
| ShareGPT4Video [10] | Llama3-8B | ∼1.0M | 43.2 | 48.7 | 27.8 | 16.1 |
| LLaVA-OV (SI) [31] | Qwen2-7B | ∼7.2M | 60.3 | 61.8 | 31.4 | 36.4 |
| LLaVA-OV [31] | Qwen2-7B | ∼8.8M | 48.0 | 71.7 | 33.2 | 45.6 |
| LLaVA-Video [104] | Qwen2-7B | ∼7.4M | 45.1 | 67.0 | 34.1 | 53.2 |
| InternVL2 [13] | InternLM2.5-7B | N/A | 58.6 | 66.5 | 39.8 | 45.5 |
| Qwen2-VL-Instruct [82] | Qwen2-7B | N/A | 48.3 | 64.9 | 38.2 | 59.4 |
| Qwen2-VL-Instruct [82] | Qwen2-72B | N/A | 55.5 | 74.7 | 45.5 | 74.6 |
| *Our models* | | | | | | |
| LLaVA-NeXT-INST-IT | Vicuna-7B | 920K | 68.6 | 63.0 | 49.3 | 42.1 |
| LLaVA-NeXT-INST-IT | Qwen2-7B | 920K | 67.9 | 75.3 | 45.7 | 53.3 |

## 3 Experiments

### 3.1 Implementation details

We use LLaVA-NeXT [44] as our baseline due to its widespread adoption. In the default configuration, Vicuna-1.5-7B [16] serves as the language model with CLIP-ViT-336 [67] as the vision encoder. We utilize the AdamW [49] with a cosine learning rate schedule for optimization. During the vision-language alignment stage, we use the LCS-558K dataset [43], and for the supervised fine-tuning stage, we leverage the open-source LLaVA-NeXT-DATA [48]. For single images, we split the original image into up to 4 sub-images based on its aspect ratio following the AnyRes [44] approach, and then concatenate the global image with these sub-images. For multiple images and video inputs, we skip the AnyRes procedure and encode every single image. Additionally, we apply $2 \times 2$ spatial pooling to reduce the number of visual tokens for video inputs. We limit the maximum number of frames to 32 and the context length of LLMs to 6K due to GPU memory constraints. To enhance instance-level understanding with our INST-IT Dataset, we combine INST-IT Dataset with LLaVA-Next-DATA in an additional continuous supervised fine-tuning stage. In this stage, we freeze the first 12 layers of the vision encoder to mitigate potential distribution shifts caused by visually prompted images. Furthermore, we use Qwen2-7B [87] with SigLIP-SO400M-384 [97] for improved performance in our main experiment, and Qwen2-1.5B with CLIP-ViT-336 for efficiency in our ablation study. We use $8 \times$H100 for all experiments. The image-video joint training stage takes approximately 20 hours when using Vicuna-7B as the language model and 24 hours using Qwen2-7B with SigLIP-SO400M-384.

### 3.2 Main experiments

**Results on INST-IT Bench.** We conduct extensive evaluations on INST-IT Bench. The results in Tab. 2 show that with instruction tuning using INST-IT Dataset, our models achieve a significant improvement of nearly 20% on average score, validating the effectiveness of INST-IT. Moreover, although ViP-LLaVA [5] utilizes visual prompts for instruction tuning, it shows minor improvement over its baseline, *i.e.* LLaVA-1.5 [43], possibly due to overfitting to its training data. In contrast, our model demonstrates consistent improvements on other instance understanding benchmarks, such as ViP-Bench [5] and RefCOCOg [53] (Sec. 3.3), as well as on general-purpose evaluation sets like AI2D and Egoschema (will be discussed in the following sections). This suggests that the model

Table 3: **Main results on image benchmarks.**

| Method | LLM | Vision Encoder | AI2D[28] (test) | MMMU[95] (val) | POPE[37] (test F1) | GQA[26] (val) | ChartQA[54] (test) |
|---|---|---|---|---|---|---|---|
| LLaVA-1.5 [43] | Vicuna-7B | CLIP-ViT-Large | 54.8 | 35.3 | 85.9 | 62.0 | 18.2 |
| DeepStack-L [60] | Vicuna-7B | CLIP-ViT-Large | - | 35.7 | 86.7 | 63.1 | 21.0 |
| DeepStack-L-HD [60] | Vicuna-7B | CLIP-ViT-Large | - | 35.6 | 86.5 | 65.2 | 56.3 |
| VILA [42] | Vicuna-7B | CLIP-ViT-Large | - | - | 85.5 | 62.3 | - |
| LLaVA-OV (SI) [31] | Qwen-7B | SigLIP-SO400M | 81.6 | 47.3 | - | - | 78.8 |
| LLaVA-OV [31] | Qwen-7B | SigLIP-SO400M | 81.4 | 48.8 | - | - | 80.0 |
| Qwen2-VL-Instruct [82] | Qwen-7B | DFN-CLIP-H | 83.0 | 54.1 | - | - | 83.0 |
| LLaVA-NeXT [44] (baseline) | Vicuna-7B | CLIP-ViT-Large | 66.6 | 35.1 | 86.4 | 64.2 | 54.8 |
| LLaVA-NeXT-INST-IT (ours) | Vicuna-7B | CLIP-ViT-Large | 71.0 ↑4.4 | 37.4 ↑2.3 | 87.2 ↑0.8 | 65.9 ↑1.7 | 68.3 ↑13.5 |
| LLaVA-NeXT-INST-IT (ours) | Qwen2-7B | SigLIP-SO400 | 78.7 ↑12.1 | 42.7 ↑7.6 | 87.6 ↑0.2 | 65.5 ↑1.3 | 72.8 ↑18.0 |

Table 4: **Main results on video benchmarks.** We report the average of MCQA, Y/N and CM in TempCompass for determinism results. * indicates results reproduced by us.

| Method | LLM | Vision Encoder | ANetQA[94] (oe) | EgoSchema[52] (subset) | NExTQA[85] (mc) | VideoMME[20] (w/o subs) | TempCompass[46] (3 avg) |
|---|---|---|---|---|---|---|---|
| DeepStack-L [60] | Vicuna-7B | CLIP-ViT-Large | 49.3 | 38.4 | 61.0 | - | - |
| Video-ChatGPT [51] | Vicuna-7B | CLIP-ViT-Large | 35.2 | 47.3 | - | - | - |
| VideoLLaMA2 [14] | Vicuna-7B | CLIP-ViT-Large | 50.2 | - | 51.7 | - | - |
| LLaVA-Next-Video [103] | Vicuna-7B | CLIP-ViT-Large | 53.5 | 43.9 | - | 46.5 | - |
| InternVL2 [13] | InternLM-7B | InternViT-300M | - | - | - | 54.0 | - |
| LLaVA-OV [31] | Qwen2-7B | SigLIP-SO400M | 56.6 | 60.1 | 79.4 | 58.2 | 69.4 |
| LLaVA-Video [104] | Qwen2-7B | SigLIP-SO400M | 56.5 | 57.3 | 83.2 | 63.3 | - |
| Qwen2-VL-Instruct [82] | Qwen2-7B | DFN-CLIP-H | - | 66.7 | - | 63.3 | 72.9 |
| LLaVA-NeXT [44] (baseline) | Vicuna-7B | CLIP-ViT-Large | 53.8 | 50.0* | 58.4* | 36.2* | 56.8* |
| LLaVA-NeXT-INST-IT (ours) | Vicuna-7B | CLIP-ViT-Large | 53.7 ↓0.1 | 57.8 ↑7.8 | 70.2 ↑11.8 | 44.3 ↑8.1 | 59.8 ↑3.0 |
| LLaVA-NeXT-INST-IT (ours) | Qwen2-7B | SigLIP-SO400 | 55.2 ↑1.4 | 50.4 ↑0.4 | 73.0 ↑14.6 | 54.0 ↑17.8 | 63.9 ↑7.1 |

trained with INST-IT generalizes well to other tasks. Qwen2VL-72B does not show substantial improvements over its smaller 7B model, indicating that simply scaling up the model size cannot address the challenges in instance understanding. Similarly, by comparing the amount of instruction tuning data used by each model, we observe that large-scale coarse-grained annotations do not lead to essential improvements either. This highlights the importance of instance-specific annotated data.

**Results on generic benchmarks.** To evaluate general understanding capabilities, we assess our models on several widely used image and video benchmarks using the LMMs-Eval [100]. To ensure a fair comparison with other models, we primarily report results from their original papers or reproduced results in previous studies. On generic image benchmarks, as shown in Tab. 3, INST-IT consistently outperforms our direct baseline model, *i.e.* LLaVA-NeXT. The improvement in AI2D, a benchmark that requires grounding and referring understanding capability, is particularly clear. This suggests that INST-IT effectively boosts the model in fine-grained understanding. Furthermore, when utilizing a more advanced language model and vision encoder, our method achieves performance comparable to large-scale SFT LMMs, such as LLaVA-OV and Qwen2-VL-Instruct, **while requiring significantly less computational and data cost.** For video understanding benchmarks in Tab. 4, INST-IT significantly outperforms both LLaVA-NeXT and LLaVA-NeXT-Video. These consistent improvements demonstrate that enhancing instance-level understanding through explicit visual prompted instruction tuning is an effective strategy for improving generic spatiotemporal understanding capabilities.

## 3.3 Evaluation on other instance-understanding benchmarks

To assess whether our model has learned generalizable instance understanding capability, we conducted evaluations on out-of-domain instance understanding benchmarks in **zero-shot** manner.

**ViP-Bench [5]** is a region-level understanding benchmark that closely aligns with the objectives of INST-IT. As shown in Tab. 5, our model exhibits strong generalization performance. In particular, our INST-IT with Vicuna-7B achieves performance comparable to ViP-LLaVA when using rectangular bounding boxes as visual prompts and even surpasses ViP-LLaVA when employing human-style visual prompts. Notably, our model performs as a generalist under **zero-shot** evaluation, whereas ViP-LLaVA benefits from in-domain tuning, since it is fine-tuned on the dataset of ViP-Bench.

Table 5: **Results on ViP-Bench.** We perform evaluation with our INST-IT models without fine-tuning.

| Model | Synthesized visual prompts | | | | | | | Visual prompts from human | | | | | | |
| | Rec | OCR | Know | Math | Rel | Lang | **All** | Rec | OCR | Know | Math | Rel | Lang | **All** |
|---|---|---|---|---|---|---|---|---|---|---|---|---|---|---|
| GPT-4V-turbo-detail:high [1] | 58.1 | 69.8 | 59.5 | 71.0 | 61.4 | 51.9 | 60.7 | 56.9 | 69.7 | 63.7 | 80.6 | 61.1 | 45.6 | 59.9 |
| GPT-4V-turbo-detail:low [1] | 53.2 | 50.3 | 55.6 | 67.7 | 57.5 | 57.5 | 52.8 | 51.7 | 50.3 | 59.3 | 60.3 | 55.0 | 43.8 | 51.4 |
| InstructBLIP-7B [17] | 36.9 | 16.3 | 34.2 | 22.3 | 26.8 | 7.5 | 31.7 | 38.9 | 17 | 35.4 | 9.7 | 29.3 | 17.5 | 33.3 |
| Shikra-7B [8] | 40.2 | 10.0 | 28.0 | 3.5 | 18.9 | 20.6 | 33.7 | – | – | – | – | – | – | – |
| GPT4ROI-7B [101] | 35.6 | 16.7 | 29.7 | 9.7 | 32.5 | 13.8 | 35.1 | – | – | – | – | – | – | – |
| Kosmos-2 [63] | 29.5 | 14.2 | 18.5 | 9.7 | 7.5 | 21.9 | 26.9 | – | – | – | – | – | – | – |
| LLaVA-1.5-7B [43] | 50.8 | 12.4 | 49.2 | 6.5 | 51.8 | 23.8 | 41.6 | 49.1 | 13.0 | 42.9 | 9.7 | 50.0 | 27.5 | 40.2 |
| Qwen-VL-Chat [4] | 43.0 | 30.4 | 40.2 | 9.7 | 25.7 | 28.7 | 39.2 | 48.7 | 22.1 | 41.2 | 6.5 | 48.2 | 25.0 | 41.7 |
| ViP-LLaVA-7B [5] | 54.8 | 18.8 | 52.9 | 9.7 | 53.9 | 42.5 | 45.5 | 55.3 | 17.6 | 45.9 | 8.1 | 44.6 | 33.1 | 46.8 |
| LLaVA-NeXT-INST-IT-Vicuna-7B | 51.3 | 23.7 | 54.2 | 12.9 | 64.3 | 46.2 | 45.1 | 55.0 | 21.3 | 52.5 | 16.1 | 57.5 | 40.6 | 48.2 |
| LLaVA-NeXT-INST-IT-Qwen2-7B | 58.9 | 24.5 | 48.5 | 12.9 | 48.2 | 46.3 | **50.5** | 57.7 | 22.5 | 53.2 | 19.4 | 53.6 | 45.0 | **49.0** |

**RefCOCOg [53]** is a referring expression comprehension benchmark, with fewer labeling errors than its counterpart RefCOCO [27]. We evaluate our LLaVA-NeXT-INST-IT-Vicuna-7B model on this benchmark and observe a clear improvement of 10.8% over the baseline LLaVA-NeXT-Vicuna-7B (63.0% vs. 52.2%). This further confirms that our approach effectively enhances the model in instance understanding, rather than simply overfitting to our INST-IT data format.

## 3.4 Ablation study

We use Qwen2-1.5B [87] as the language model and CLIP-ViT-L-336 [68] as the vision encoder for ablation experiments. We first conduct ablation on the training recipe to investigate how to effectively integrate INST-IT Dataset with existing academic SFT datasets [48] for a balanced improvement. Next, we perform a detailed analysis of the impact of each component in our INST-IT Dataset.

**Effectiveness of our continuous instruction-tuning paradigm.** As shown in Tab. 6, directly mixing the video split of INST-IT Dataset with LLaVA-Next-DATA leads to significant improvements on video benchmarks. However, the performance on generic image understanding slightly declines. We believe this is due to two main reasons: (1) the increased ratio of video data may suppress image understanding; (2) visually prompted images may introduce a distribution shift from natural images. To address these issues, we propose a continuous SFT paradigm based on single-image models and freeze the first 12 layers of the vision encoder to preserve realistic low-level features. Our model achieves balanced performance across both image and video benchmarks with this training approach.

**Detailed dataset combination.** As illustrated in Fig. 2, INST-IT Dataset contains fine-grained annotations at multi-level. To investigate the effectiveness of each component in INST-IT Dataset, we conduct an extensive ablation by progressively adding data components. As shown in Tab. 7, the instance-level and image-level frame captions are essential for improving instance understanding in images. Meanwhile, temporal differences, along with video-level descriptions and QA, significantly enhance video instance understanding. Finally, incorporating the image component of INST-IT Dataset enables our model to achieve the most balanced performance across generic image and video understanding benchmarks, as well as our INST-IT Bench.

## 4 Related Work

**Large multimodal models.** Recently, significant progress has been witnessed in LMMs [91]. BLIP-2 [34] and Flamingo [2] leverage visual re-samplers to integrate image features as language inputs by extracting a fixed number of visual tokens. LLaVA [45] and its follow-ups [43, 31, 42, 57, 98, 60, 11] achieve remarkable success by connecting vision and language through a simple projection module. Additionally, researchers are extending LMMs' capabilities to temporal understanding by incorporating multi-frame inputs [41, 82, 104] or explicit temporal modules [39, 25] However, existing LMMs struggle with instance-level understanding and often fail to accurately follow instructions to ground specific instances. We emphasize the importance of instance understanding and enhance it through instruction fine-tuning with explicit visual prompts.

Table 6: **Ablation on data training recipe.** L.N. denotes LLaVA-NeXT-Data, while INST-IT $_{img}$ and INST-IT $_{vid}$ refer to the image and video subsets of INST-IT. INST-IT-I and INST-IT-V indicate the multi-choice splits of the image and video part of our INST-IT Bench, respectively.

| CL | Tune Enc | Data Combination | AI2D (test) | POPE (test F1) | GQA (val) | INST-IT-I (mc) | Next-QA (mc) | VideoMME (w/o subt) | INST-IT-V (mc) |
|---|---|---|---|---|---|---|---|---|---|
| | All | L.N. | 61.1 | 86.9 | 61.4 | 45.3 | 56.6 | 45.7 | 31.3 |
| | All | L.N. & INST-IT $_{vid}$ | 60.7 | 86.1 | 61.2 | 60.7 | 59.7 | 47.1 | 43.0 |
| ✓ | All | L.N. & INST-IT $_{vid}$ | 62.3 | 86.7 | 62.9 | 61.8 | 62.4 | 46.7 | 44.4 |
| ✓ | None | L.N. & INST-IT $_{vid}$ | 63.1 | 86.9 | 62.5 | 60.2 | 63.2 | 47.2 | 44.3 |
| ✓ | Last 12 | L.N. & INST-IT $_{vid}$ | 63.2 | 87.0 | 62.5 | 60.1 | 63.3 | 47.2 | 44.0 |
| ✓ | None | L.N. & INST-IT $_{img+vid}$ | 63.0 | 87.0 | 62.7 | 58.6 | 59.8 | 46.7 | 41.6 |
| ✓ | Last 12 | L.N. & INST-IT $_{img+vid}$ | 63.0 | 87.2 | 62.7 | 59.6 | 64.3 | 46.6 | 43.7 |

Table 7: **Ablation on detailed data combination.** The dataset combination in line #3 corresponds to the video part of INST-IT Dataset, while line #4 represents the complete INST-IT Dataset by incorporating the image part into line #3.

| # | Data Combination | AI2D (test) | MMMU (val) | POPE (F1) | GQA (val) | INST-IT-I (mc) | Next-QA (mc) | VideoMME (w/o subt) | INST-IT-V (mc) |
|---|---|---|---|---|---|---|---|---|---|
| 0 | LLaVA-NeXT | 61.1 | 35.9 | 86.9 | 61.4 | 45.3 | 56.6 | 45.7 | 31.3 |
| 1 | + inst-cap & img-cap | 63.0 | 35.1 | 86.1 | 62.7 | 58.9 | 62.4 | 46.0 | 33.8 |
| 2 | + temporal diff | 63.0 | 35.6 | 87.1 | 62.7 | 59.6 | 64.2 | 45.6 | 36.9 |
| 3 | + video-description & qa | 63.2 | 34.9 | 87.0 | 62.5 | 60.1 | 63.3 | 47.2 | 44.0 |
| 4 | + INST-IT Dataset $_{img}$ | 63.0 | 36.1 | 87.2 | 62.7 | 59.6 | 64.3 | 46.6 | 43.7 |

**Multimodal datasets and benchmarks.** With the rapid progress in LMMs, numerous instruction-tuning datasets have been developed. LLaVA-Instruct [45] leverages object categories, bounding boxes, and image-level captions to generate diverse visual instruction tuning data. Follow-up studies use more powerful models to generate synthetic data [9, 81, 7] and improve the annotation pipeline [36, 10, 104]. Simultaneously, various benchmarks are proposed to evaluate LMMs across different aspects [21, 35, 40], such as comprehensive understanding [30], OCR [54, 56, 55, 78], temporal understanding [20, 52, 85, 6, 46, 47], and instruction-following [66]. However, they focus more on image or video-level understanding and lack fine-grained emphasis on specific instances. We emphasize the importance of instance understanding in both images and videos, and propose the INST-IT Bench to evaluate the instance understanding of LMMs and create the INST-IT Dataset, providing detailed instance-level annotations to enhance instance understanding.

**Multimodal instance understanding.** Understanding individual instances is a central focus in computer vision community, with key tasks like object detection [73, 71, 12], instance segmentation [74, 29], and object tracking [19, 50, 90]. In the era of LMMs, instance understanding gains increasing attention. SPEC [62], ARO[96], and Winoground [77] reveal that CLIP [68] struggle to understand instances. To address this, KOSMOS-2 [64], Ferret [92], GLaMM [69] and Shikra [8] encode instance information in textual form. In parallel, SoM-LLaVA [86], RegionGPT [24], GPT4ROI [101], MG-LLaVA [105], OMG-LLaVA [102], and ViP-LLaVA [5], explores the use of visual prompting to guide models in focusing on specific instances. SoM-LLaVA [86] and Elysium [80] are closely related to ours. SoM-LLaVA [86] asks models to list the instances in images, finding this effective in enhancing model comprehension. However, it is limited to the image domain. Elysium [80] focuses on object understanding in videos but employs relatively simplistic instance annotations. In contrast, we focus on both images and videos and provide multi-level fine-grained annotations for instances, aiming to advance multimodal models in understanding the spatiotemporal dynamics of individual instances.

## 5   Conclusion

Instance understanding that detects, segments, and reasons nuanced relationships among objects has long been the goal of computer vision research, yet limited effort has been made to equip LMMs

with such capabilities. We introduced INST-IT Bench, a carefully curated benchmark for evaluating multimodal instance understanding abilities. Extensive evaluations for a wide range of models demonstrate the limitations of current models for understanding at the instance level. To mitigate this issue, we collected INST-IT Dataset, the first instruction-tuning dataset with explicit instance-level visual prompts and annotations. Based on INST-IT Dataset, we proposed INST-IT, a continuous finetuning framework that excels in instance understanding and general comprehension.

**Acknowledgement** This work was supported in part by the National Natural Science Foundation of China (Grant 62472098) and the Science and Technology Commission of Shanghai Municipality (No. 24511103100).

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

# Appendix

- In Sec. A, we outline additional implementation details of the GPT-4o-assisted data annotation pipeline.
- In Sec. B, we present further information about the instance understanding benchmark, INST-IT Bench.
- In Sec. C, we share more details about the instruction fine-tuning dataset, INST-IT Dataset.
- In Sec. D, we provide more discussions on failure cases and real-world applications.

## A   Data Annotation Pipeline

### A.1   Set-of-Marks Visual Prompting

Performing instance-level annotations is challenging, and we adopt the SoM visual prompting technique [88] to address this. Specifically, as illustrated in Fig. 3, we overlay a numeric ID at the center of each instance and maintain the same ID for a given instance across all frames. This simple augmentation can explicitly guide GPT-4o to focus more effectively on the instances of interest, enabling finer-grained and more accurate annotations. Furthermore, segmentation masks are necessary to calculate the center coordinates of each instance. Details on how these masks are obtained are provided in Sec. C.1.

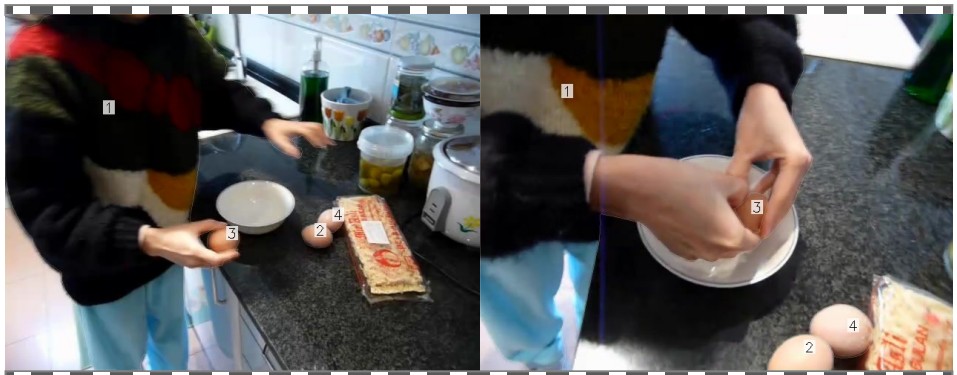

Figure 3: **Set-of-Marks visual prompting on the original videos.** Each instance is assigned a unique numeric ID, which remains consistent across all frames.

### A.2   Prompting GPT-4o

**Task prompt templates.** Prompt engineering is crucial for enabling GPT-4o to accomplish specific tasks. In this section, we present the task prompts that we designed to prompt GPT-4o for data annotation:

- The task prompt $P^f$ for frame-level annotation, Fig. 5.
- The task prompt $P^{vid}$ for video-level annotation, Fig. 6.
- The task prompt $P^{qa}$ for open-ended question-answer pairs generating, Fig. 7.

**GPT-4o API version.** During the annotation process, we use the GPT-4o-2024-08-06 API and leverage its structured output functionality to facilitate output parsing, enabling the model to respond in a predefined JSON format.

## B   More Details about INST-IT Bench

### B.1   Negative Options Generation

We use the ground-truth from open-ended QA as the positive option and additionally craft three negative options, forming a multiple-choice question with four options. To create hard negatives, we

Figure 4: **GPT-4o-based open-ended question answering correctness assessment.** The underlined parts in the figure are included only when evaluating the video split, while the *italicized* parts will be replaced by the actual sample for scoring.

Figure 5: **Frame-level annotation task prompt**, the *italicized* part are placeholders for the actual inputs.

**# Task Description:**
You are an expert in summarizing video content. Given a sequence of frame-by-frame text descriptions of a video. Your task is to aggregate these descriptions into an accurate, cohesive summary of the entire video.

**# Guidelines and Rules:**
- Base your description solely on the input to ensure accuracy; avoid inferring any unmentioned content.
- Please note that the description of a single frame may contain some inaccuracies. You need to use the overall context to further correct these errors, ensuring accuracy and consistency.
- Use chronological order: organize your summary according to the timestamps of the frames, follow these conventions: for specific moments, write <timestamp>, e.g., at <3>; for time intervals: write <start_timestamp>-<end_timestamp>, e.g., during <5>-<7>
- Referencing objects by ID: in your response, use the same [ID] format provided in the input to reference objects: for one object: [ID] (e.g., [8] a white dog); for multiple objects: [ID1] [ID2] ... (e.g., [3] [4] [5]).

**# Output Requirements:**
Your output should be a dense, detailed, and accurate description of the entire video, summarizing main objects, key events, and various spatial and event-related details.

**# Input Format:**
Each frame's description includes four parts:
1. Timestamp: marks the chronological position of the frame in the video.
2. Instance-level description: lists the primary objects in the frame using the format "[object ID]: object description"
3. Frame-level description: offers a comprehensive view of the frame's content, covering main objects, object relationships, and the background or environment details.
4. Temporal change description: highlights key changes or movements since the previous frame, capturing dynamic information essential for understanding the video's progression.

**# Input Frame-level Annotations:**
*Timestamp: <1>; Instance-level description: ... ; Frame-level description: ... ; Temporal changes: None, as this is the first frame.*
*Timestamp: <2>; Instance-level description: ... ; Frame-level description: ... ; Temporal changes: ...*
*...*

Figure 6: **Video-level annotation task prompt**, the *italicized* part are placeholders for the actual inputs.

**# Task Description :**
You are an expert in video content analysis. In this task, you will receive textual descriptions of individual video frames. Your task is to generate high-quality and contextually coherent questions and accurate answers based on the content of the video.

**# Guidelines:**
- Avoid speculative questions; ensure all questions can be answered from the frame descriptions.
- Diversify the types of questions (who, what, where, when, how, why) to cover different aspects of the video.
- The number of question-answer pairs should between 10 to 20, this depends on how much valuable information contained in the video.
- Be creative and flexible in forming questions and answers, and avoid redundant or overly simple questions.
- Use the frame timestamps to express time in the video: for a specific moment, use <timestamp> , e.g., at <3>; for a time interval, use <start_timestamp>-<end_timestamp>, e.g., during <5>-<7>. Don't forget to enclose the timestamps in <>.
- In the input, ID is used to refer to a specific object; you can use the same format in your output to refer to specific objects: for a single object, write [ID] (e.g., "[8]"); for multiple objects, use "[ID1] [ID2] ...", such as "[3] [4] [5]".

**# Output:**
The output is a list of 10 to 20 high-quality, context-aware question-answer pairs about the video's content.

**# Input Format:**
The input consists of frame-by-frame descriptions, where each frame includes:
1. Timestamp: marks the chronological position of the frame in the video.
2. Frame-level description: offers a comprehensive view of the frame's content, covering main objects, object relationships, and the background or environment details.
3. Temporal change description: highlights key changes or movements since the previous frame, capturing dynamic information essential for understanding the video's progression.

**# Input Frame-level Annotations:**
*Timestamp: <1>; Frame-level description: ... ; Temporal changes: None, as this is the first frame.*
*Timestamp: <2>; Frame-level description: ... ; Temporal changes: ...*
*...*

Figure 7: **Open-ended question-answer pairs generation task prompt**, the *italicized* part are placeholders for the actual inputs.

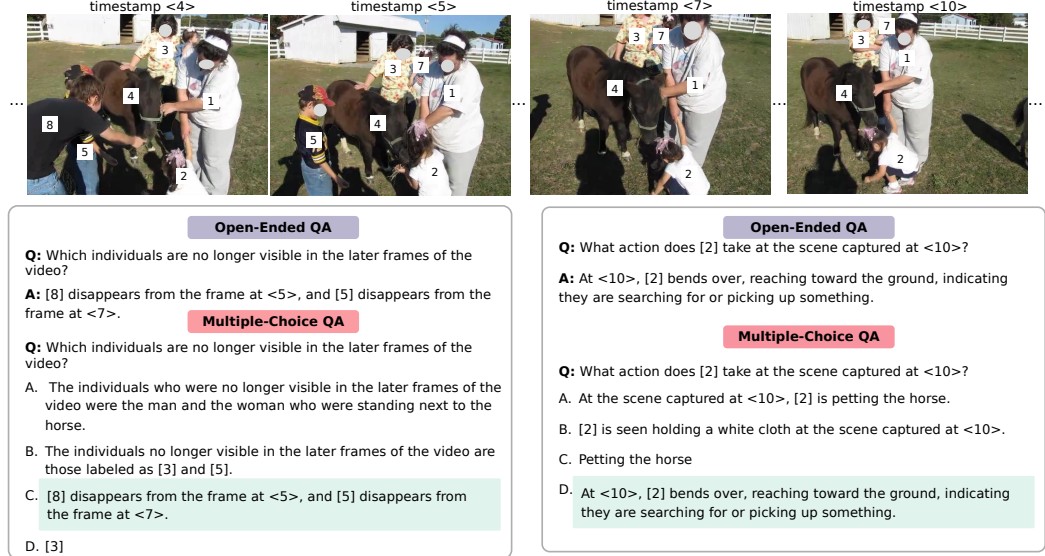

Figure 8: **A data example from INST-IT Bench.** Each test sample includes both open-ended QA and multiple-choice QA, focusing on specific instances or the relationships and interactions between instances.

first have the model answer the open-ended questions and use GPT-4o to score the correctness of the responses. If the score is lower than 0.4, we consider it a difficult negative answer and include it as one of the negative options. Finally, we randomly shuffle the four options to ensure that the correct one appears in each position with equal probability.

## B.2 LLM-based Evaluator for Open-Ended QA

Recent studies [93, 15] suggest that LLMs can serve as effective evaluators. Building on this, we use GPT-4o to assess the accuracy of open-ended question answering. Specifically, GPT-4o assigns a score between 0 and 1 based on three key factors: the question, the ground-truth answer, and the model prediction. Given that INST-IT Bench prioritizes instance-level understanding, we pay special attention to the accuracy of instance ID references. Furthermore, for the video split of INST-IT Bench, we emphasize the correctness of timestamps to ensure temporal correctness. The task prompt for GPT-4o is illustrated in Fig. 4.

## B.3 Data Example

To provide a clearer understanding of INST-IT Bench, we present a data example in Fig. 8. Each question includes both open-ended and multiple-choice formats, focusing on specific instances or exploring the relationships and interactions between multiple instances. This design highlights the significant distinction from other benchmarks, emphasizing fine-grained understanding at the instance level.

## C More Details about INST-IT Dataset

### C.1 Data Collection and Processing

**Collection.** We select five instance segmentation datasets and two multi-object tracking datasets as sources of video data. To prevent data leakage, we only used the training splits of these datasets, leaving their test and validation splits untouched. Additionally, we use the SA-1B [29] dataset as a source of image data and only utilize the first ten officially provided data splits. For each split, we

Table 8: **Data sources.** We use seven video datasets and one image dataset as our data sources. We show their annotation formats, the splits we used, and the number of samples from each dataset.

| Dataset Name | Ann. Type | Split | Sample Num. |
|---|---|---|---|
| *Video Instance Segmentation* | | | |
| BRUST [3] | mask | training | 500 |
| UVO [83] | mask | training | 5,135 |
| OVIS [65] | mask | training | 599 |
| LVVIS [79] | mask | training | 3,057 |
| YoutubeVIS [89] | mask | training | 2,897 |
| *Video Object Tracking* | | | |
| BenSMOT [38] | box | training | 2,261 |
| VidOR [75] | box | training | 6,969 |
| *Image* | | | |
| SA-1B [65] | none | 1-10 | 51,101 |

only use the first 50% of its images. In total, we collect 21,418 videos and 51,101 images. Tab. 8 provides detailed statistics on our data sources.

**Processing.** When constructing SoM [88] visual prompts, we need to obtain the mask annotations for each instance to determine the location of the numeric IDs. For the video instance segmentation datasets [3, 83, 65, 79, 89], the instance masks are already provided and can be used directly. For multi-object tracking datasets [38, 75], we prompt SAM [29] with their bounding box annotations to generate instance masks. For images in the SA-1B dataset [29], we employ Semantic-SAM [33] to segment the instances and obtain their masks.

### C.2 Statistics Analysis.

**Number of instances.** The key characteristic of INST-IT Dataset is its specific focus on individual instances in images and videos, which provides a more fine-grained description of the visual inputs. We visualize the distribution of the number of instances in each sample in Fig. 9. For the video split, each sample has an average of 3.7 instances, with a total of 79,709 instances. For the image split, each sample contains an average of 6.9 instances, totaling 351,495 instances. Across the entire dataset, each sample includes an average of 5.9 instances, adding up to 431,204 instances in total. We measure the scene complicity by the number of instances in each sample. Specifically, 31% of the samples contain $\leq 3$ instances (simple), 39% have between 3 to 8 instances (medium), and the remaining 30% contain $\geq 8$ instances (hard).

**Dataset diversity.** We visualize the object categories in INST-IT Dataset in Fig. 10, highlighting its diverse range. The objects include humans, animals, plants, vehicles, landmarks, etc. , covering domains like daily life, egocentric perspectives, sports, transportation, etc. . The rich diversity of INST-IT Dataset ensures its applicability to real-world scenarios and enhances its transferability to different domains.

**Text captions.** INST-IT Dataset contains multi-level textual descriptions of visual content, covering instances, frames, temporal changes, and video-level annotations. We conduct statistical analysis on these text annotations, including the number of each type of text, and their average length. As shown in Tab. 9, the average length of INST-IT Dataset is 49.1 words per caption, with video-level averaging 323.2 words, highlighting its richness of details. We also present the results of lexical analysis in Tab. 9. The instance-level captions contain a rich variety of nouns and adjectives, indicating that they primarily describe the objects' categories and attributes. The captions of temporal changes include a high volume of verbs and adverbs, suggesting that they capture dynamic information.

**Human evaluation of data quality** We invited three volunteers to rate each sample on a scale from 1 to 5, with higher scores indicating better quality. Tab. 10 presents the scores of different types of annotations, along with the average time spent by each volunteer to evaluate each sample. The average score across all types is $4.49_{\pm 0.05}$, indicating that the data in INST-IT Dataset is of satisfactory quality.

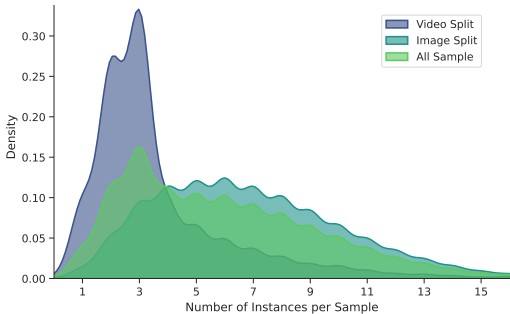

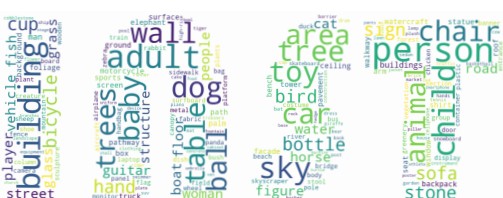

Figure 9: **The distribution of the number of instances per sample in INST-IT Dataset.** We separately present the distribution for the video split, image split, and the entire dataset.

Figure 10: **Analysis of object categories in INST-IT Dataset**, which shows a diverse range of types spanning multiple domains.

Table 9: **Statistical and lexical analysis of INST-IT Dataset.** We present the results for each annotation level as well as the entire dataset.

| Caption Type | #Caption | #Char./Cap. | #Word/Cap. | #Sen./Cap. | Nouns | Adj. | Adv. | Verb. | Prep. |
|---|---|---|---|---|---|---|---|---|---|
| Instance-level | 836,524 | 102.1 | 24.3 | 1.5 | **26.5%** | **13.3%** | 2.3% | 12.3% | 10.7% |
| Frame-level | 207,662 | 458.0 | 106.5 | 5.7 | 25.2% | 10.5% | 2.6% | 14.9% | 11.5% |
| Temporal-change | 135,143 | 306.6 | 67.7 | 3.7 | 21.2% | 10.0% | **6.0%** | **16.4%** | 10.8% |
| Video-level | 21,372 | 1441.8 | 342.2 | 14.3 | 24.8% | 10.6% | 3.6% | 13.2% | **11.8%** |
| All | 1,200,701 | 210.5 | 49.1 | 2.7 | 25.0% | 11.4% | 3.1% | 14.0% | 11.1% |

## C.3 Data example.

In this section, we provide a complete video data sample from INST-IT Dataset to offer a clearer understanding of its content and format. In all annotations, we use the format [ID] to refer to instances and <timestamp> to refer to timestamps. We present the frame-level annotations in Tab. 11. We can see that each frame-level annotation $Y^f$ consists of three parts: instance-level descriptions $y^{ins}$, image-level descriptions $y^{img}$, and temporal differences $y_{dif}$. Additionally, each video is accompanied by a series of open-ended question-answer pairs $Y^{qa}$, most of which center on specific instances or their relationships, as illustrated in Tab. 12. Furthermore, we generate a dense video-level caption $Y^{vid}$ summarizing the entire video in chronological order, as shown in Tab. 13.

## D More discussions.

### D.1 Failure cases.

We manually inspect the dataset and model to identify the failure cases. We find that occasional failures occur in scenarios where instances are severely occluded, the image is blurry, or instances are excessively small or crowded. These challenges are common among LMMs, and future research can further investigate them.

### D.2 Real-world applications.

In real-world applications, users can interactively prompt models like SAM2 [70] to automatically track instances of interest and generate SoMs. Additionally, our model also supports inputs without SoMs, allowing users to specify particular instances using textual descriptions. In the first scenario, our INST-IT introduces only a marginal overhead for generating SoMs, while in the second case, it incurs no extra cost compared to the base model.

## E Limitations and broader impacts.

**Limitations.** Our current experiments are conducted on 7B and 1.5B models due to the computation cost. Moreover, our current data pipeline is automated but constrained by the overhead of GPT-4o.

Table 10: Human evaluation on the quality of INST-IT Dataset.

| | Instance Caption | Image Caption | Temporal Caption | Video Caption | QA Pairs |
|---|---|---|---|---|---|
| Score ($\uparrow$) | $4.66_{\pm 0.12}$ | $4.68_{\pm 0.02}$ | $4.48_{\pm 0.05}$ | $4.34_{\pm 0.18}$ | $4.31_{\pm 0.11}$ |
| Time (s) | 7.3 | 12.4 | 11.9 | 31.0 | 10.6 |

We can further scale the model size and scale the dataset using a model-in-the-loop approach and improve the model through multi-round instruction tuning with self-synthesized data. We leave this direction for future work.

**Broader impacts.** This paper proposes an enhancement of instance-level understanding capabilities in large multimodal models, enabling them to better assist users by answering questions about the content of interest. However, similar to existing large multimodal models, this approach also faces potential risks, such as issues related to fairness and bias. Future work can address this issue through approaches such as data filtering and validation.

Table 11: **INST-IT Dataset frame-level annotations**. For the ease of visualization, we only demonstrate the first three frames. Please zoom in to view the instance ID labels.

| Frame | Instance-level captions | Image-level captions | Temporal differences |
|---|---|---|---|
| 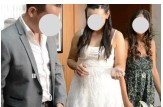 timestamp<1> | 1: Wearing a light gray suit with a white shirt, standing indoors. 2: Wearing a sleeveless white lace dress, holding an object in the hand. 3: Wearing a dark floral-patterned dress with long wavy hair. | [1] [2] [3] are standing closely together in an indoor setting. [1] is on the left side wearing a formal, light gray suit with a white shirt. [2], in the middle, is wearing a sleeveless white lace dress, holding something in their hand. [3] is on the right side in a dark floral-patterned dress with long, wavy hair. They appear to be in a room with wooden paneling and some framed art on the wall. null | |
| 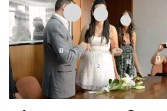 timestamp<2> | 1: A person wearing a gray suit with a white shirt, short hair. 2: A person in a white, sleeveless dress with long dark hair. 3: A person wearing a dark floral dress with long dark hair. 5: A person wearing red, partially visible in the background. 6: A small black cellphone held in a hand. | The scene appears to be in an office setting with a wooden table at the foreground. [1] is standing to the left, facing [2], and appears to be holding [2]'s finger or hand. [2] stands slightly to the right, returning focus with [1]. [3] is to the right of [2], slightly in the background, smiling and looking forward. A bouquet of white flowers lies on the table near [2]. [5] is partially visible in the background on the right, seated and wearing red. [6] is a cellphone held by [5]. Background shows a wooden wall and a reflection in a window. | [1] has moved closer to [2] and is now in contact with [2]'s hand. [2] has turned slightly towards [1] compared to the previous frame. [3] remains in a similar position, but the expression suggests more engagement with the scene. [5] and [6] have appeared in the frame; [5] is visible in the background holding [6]. The table with a bouquet of flowers is now visible, indicating a shift in camera angle slightly to include more of the right side of the room. |
| 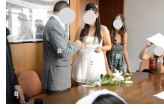 timestamp<3> | 1: Wearing a grey suit, standing beside [2] and slightly turned towards them. 2: Wearing a white, sleeveless dress with floral textures. Holding a bouquet of white flowers. 3: Wearing a dark patterned dress, standing slightly behind [2]. 4: Partially visible, wearing dark clothing, located at the edge of the left side of the frame. 5: Seated, wearing a red outfit. Holding a white object above their head, possibly obscuring their face. | The scene shows [1] [2] [3] near a wooden conference table in a professional setting, possibly an office. [1] wears a grey suit and is standing to the left, engaged with [2] who is wearing a white dress and holding flowers. [3], who is in a patterned dress, stands closely behind [2]. The newly appeared [4] is seated to the far left, partially visible at the edge of the frame. [5] is seated on the right side, holding an object above their head, possibly obscuring their face. The room has wooden walls and a framed picture hanging on the wall. | Object [5] has lifted an object above their head, possibly a piece of paper. Object [4] has appeared in the scene, seated on the left side of the frame, which was not visible earlier. The positions of objects [1], [2], and [3] remain unchanged, as does the background and setting of the room. Overall, no significant movement is noticed in terms of camera angle or position for objects [1] [2] [3]. |

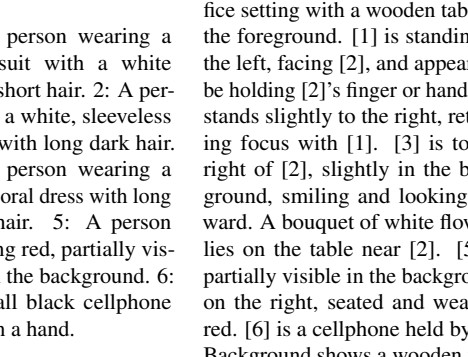

Table 12: **INST-IT Dataset Open-ended question-answer pairs.**

| Question | Answer |
| --- | --- |
| What change occurs with [1]'s expression between <10> and the previous frame? | [1] changes from smiling to a neutral expression. |
| What activity are [1] and [2] involved in at <11>? | [1] and [2] are engaged in a kiss. |
| What is the overall mood during <11> as suggested by [3]'s actions? | A celebratory or joyous event. |
| What interaction occurs between [1] and [2] at <5>? | [1] holds [2]'s hand, suggesting an intimate gesture or exchange, likely a ring. |
| Who joins [1] and [2] in the frame at <7>? | [4] appears in the frame, joining [1] and [2]. |
| What changes in the group's composition between <7> and <8>? | [3] reappears, and [4] is no longer visible. |
| What element is seen throughout the frames <1> to <12>? | The scene is in an indoor setting with wooden paneling and framed art. |
| What type of event is likely taking place based on the atmosphere in <4> and <6>? | A formal event, possibly a wedding or official gathering. |
| What new elements are introduced in the scene at <2>? | [5] holds a cellphone in the background, partially visible. |
| What is the mood and lighting like at <6>? | The mood is formal and celebratory, with bright lighting enhancing this atmosphere. |
| What new background element appears at <7>? | There is a map or blueprint on the wall. |
| What is notable about [5]'s actions at <3>? | [5] is lifting an object above their head, possibly a piece of paper. |
| What is the setting like in <3>? | The group is gathered near a wooden conference table in a formal setting. |
| How are [1] and [2] interacting at <8>? | They are engaged in conversation or communication, indicated by body language and focus. |
| What does [1]'s expression suggest at <12>? | [1] speaks or smiles, suggesting engagement with [2] or others. |
| What shift occurs in the focus of the camera between <5> and <6>? | The camera focuses more on individuals standing together, reducing focus on the foreground objects. |
| What are [3] and [4] doing at <9>? | They clapping their hands in celebration. |
| What decorative element is visible at <2>? | A bouquet of flowers lies on the table near [2]. |
| How has the posture of [1] and [2] changed by <6>? | [1] and [2] face slightly outward, suggesting a pose for a photograph or audience. |
| What overall physical change occurs between [1] and [2] from <10> to <11>? | There's a noticeable increase in their physical interaction, enhancing emotional engagement. |

Table 13: **INST-IT Dataset video-level caption.**

| Video-level caption |
| --- |

The video appears to document a formal or celebratory event indoors, possibly a ceremony such as a wedding or official gathering, occurring in a room with wooden paneling and art or framed pictures on the wall. At the beginning, during <1>, [1] is wearing a light gray suit and stands with [2] in a sleeveless white lace dress, and [3] in a dark floral-patterned dress. The three are close together, suggesting an intimate or focused setting. The progression between <2> and <3> involves subtle changes in posture and interaction. [1] moves closer to [2], appearing to hold hands or engage in an exchange, possibly involving a ring, as indicated by a bouquet of flowers. [3] remains supportive and smiling, while [5], in red, momentarily holds an object above their head, before disappearing from view by <4>.In frames <5> to <7>, [1] and [2] maintain a close interaction, suggestive of a significant moment such as an exchange of vows or rings. They are closely observed by [3], who stands smiling nearby, while [1] and [2] occasionally adjust their positions, facing each other initially and then turning outward, which may signal transitioning from an intimate moment to posing for a photo. By <7>, [4] joins, dressed in darker attire, emphasizing the formal setting as [3] is no longer visible. Through <8> and <9>, the group dynamics change slightly with the absence of [4] and [3] entering the scene again. [1] and [2] appear to engage in a warm interaction as [3] supports them, clapping, alongside the visible hands of [4] indicating applause, marking a cheerful tone. Finally, during <10> to <12>, the focus shifts as [1] and [2] first engage in a kiss, underscoring an intimate conclusion to their ceremony. They later stand apart slightly at the center, with [1] smiling or speaking, and [2] leaning towards [1] suggestively content. Throughout, the consistent joyous mood is accentuated by [3]'s ongoing clapping and expression of joy, emphasizing shared celebration and approval from the audience captured.

