# OpenReview forum: "INST-IT: Boosting Instance Understanding via Explicit Visual Prompt Instruction Tuning"
_NeurIPS.cc/2025/Conference — NeurIPS 2025 poster_

### Official Review · Reviewer_no3B · 2025-06-29

**Clarity:** 3
**Significance:** 2
**Originality:** 2
**Rating:** 3
**Confidence:** 4

**Summary:**

This paper presents INST-IT, a comprehensive framework aimed at enhancing instance-level understanding in Large Multimodal Models through explicit visual prompting and instruction tuning. The authors propose a large-scale dataset (INST-IT Dataset) that incorporates fine-grained, instance-specific annotations for both images and videos, generated through an automated pipeline using GPT-4o to highlight target instances. Besides, they construct a benchmark (INST-IT Bench) designed to evaluate instance-level understanding in both open-ended and multiple-choice formats. The training strategy, based on continuous instruction tuning, blends general instruction data with instance-level supervision, demonstrating significant improvements not only on the proposed benchmarks but also on a variety of generic image and video understanding tasks. The results show that their method is broadly effective across domains.

**Questions:**

1. Does the use of GPT-4o for large-scale annotation introduce any detectable biases or hallucinations in the model’s final responses? How was this risk assessed or mitigated during data generation and training?

2. Could you provide results or analysis comparing your approach against stronger and more recent baseline models?

3. Could the authors clarify the typical duration of the videos in the dataset and describe the specific frame sampling strategy used? Additionally, is there a risk of missing important or key frames when sampling at fixed intervals?

4. Could the authors specify the total cost of annotating the entire dataset?

**Ethical Concerns:**

["NO or VERY MINOR ethics concerns only"]

**Final Justification:**

While I appreciate the authors' detailed rebuttal, I remain unconvinced that the technical novelty and contributions are sufficient for acceptance. I will therefore keep my original score.

**Limitations:**

See above.

**Quality:**

3

**Strengths And Weaknesses:**

**Strengths**

1. The paper addresses a pressing and relatively underexplored limitation in current LMMs, specifically their poor instance-level understanding, and offers a practical and well-justified solution.

2. The dataset construction is large-scale, multi-granular, and thoughtfully designed with visual prompts, covering both images and videos, which adds notable novelty and utility to the field.

3. The benchmark (INST-IT Bench) is carefully built and provides both openended and multiple-choice QA formats, which allow for more robust evaluation.

**Weaknesses**：

1. The baselines used for comparison are not comprehensive and are relatively outdated. Including stronger and more recent models such as the InternVL series and the Qwen2.5-VL series would strengthen the validity and credibility of the performance claims.

2. The human evaluation uses 500 randomly sampled data points, which is helpful but relatively limited given the dataset's scale in the tens of thousands. Expanding the size of the human-reviewed subset would strengthen the credibility and representativeness of the quality assessment.

3. The use of visual prompting is effective but not fundamentally novel. Some parts of the methodology may be seen as incremental extensions.

4. The potential computational overhead during inference when using visual prompts (e.g., SoMs) is not quantitatively assessed, which may limit the method’s practicality in latency-sensitive applications.

---

> ### Author Rebuttal · Authors · 2025-07-31
>
> Dear reviewer no3B,
>
> Thank you for your time and thoughtful comments. We are encouraged by your recognition of our contributions as well-justified and novel. Below, we address your concerns in detail.
>
> ---
>
> > [W1, Q2] Including stronger and more recent baseline models.
>
> Thank you for the valuable suggestion. We have conducted additional experiments using the recent Qwen2.5-VL-3B-Instruct model to further evaluate the effectiveness and generalizability of our INST-IT. The results are presented in the table below.
>
> As shown, integrating INST-IT into Qwen2.5-VL consistently improves performance across a range of benchmarks. Specifically, we observe gains on general image understanding (MMMU, GQA), general video understanding (NeXTQA, VISBench), and particularly clear improvements on instance-level understanding benchmarks (RefCOCOg, INST-IT-I, INST-IT-V). These results are consistent with our previous findings on LLaVA-NeXT, further validating the effectiveness and cross-model applicability of INST-IT.
>
>
> |            | MMMU | GQA | NeXTQA | VISBench | RefCOCOg(CIDEr) | INST-IT-I | INST-IT-V |
> |------------|------|-----|--------|----------|----------|-----------|-----------|
> | Qwen2.5VL*  | 46.4 | 60.0 | 75.9 | 27.9 | 62.6 | 65.7 |  52.1  |
> | Qwen2.5VL+INST-IT | 48.4 (+2.0) | 63.3 (+3.3) | 77.4 (+1.5) | 32.9 (+5.0) | 69.0 (+6.4) | 81.3 (+15.6) |  68.6 (+16.5)  |
>
> \* We used the lmms-eval `[1]` toolkit for model evaluation and re-evaluated Qwen2.5VL-3B-Instruct using the same inference parameters and runtime configurations as our model.
>
> > [W2] Expanding the size of the human-reviewed subset.
>
> Thank you. We have further expanded the human study by including an additional 200 data points.
> The scores and the time spent reviewing each sample are shown in the table below. Consistent with the findings from the initial subset of 500 samples, the results indicate that the annotation quality in INST-IT remains satisfactory. However, since manual evaluation is very labor-intensive, we were only able to add 200 additional samples during the rebuttal phrase. We will further scale up the human study in the revised version to improve the representativeness of the evaluated subset.
>
> |                | Instance Caption | Image Caption | Temporal Caption | Video Caption | QA Pairs |
> | :------------- | :--------------- | :------------ | :--------------- | :------------ | :------- |
> | Score (↑)      |       4.61       |     4.55      |      4.42        |      4.29     |    4.30  |
> | Time (s)       |       8.4        |     13.5      |      14.0        |     40.0      |    11.9  |
> > [W3] The use of visual prompting is effective but not fundamentally novel.
>
> While visual prompts have been used in previous work for referring to specific objects during model inference `[2]` or for training image understanding models `[3]`, to the best of our knowledge, INST-IT is the first work to directly perform grounded instruction tuning using explicit visually prompted **images and videos**, and empirically demonstrated that this approach can effectively enhance the model's **spatiotemporal** fine-grained understanding capabilities.
>
> > [W4]The potential computational overhead when using visual prompts.
>
> The SAM2 series models `[4]` can be used for automated generation of SoMs visual prompts. The table below lists the officially reported `[4]` inference speeds of different model sizes (measured on an A100 using `Torch 2.5.1` and `CUDA 12.4`).
>
> | Model                 | Size (M) | Speed (Frame Per Second, FPS) |
> |----------------------|----------|--------------|
> | sam2_hiera_tiny      | 38.9     | 91.5         |
> | sam2_hiera_small     | 46       | 85.6         |
> | sam2_hiera_base_plus | 80.8     | 64.8         |
> | sam2_hiera_large     | 224.4    | 39.7         |
>
> Compared to LMMs with billions of parameters, the overhead introduced by SoMs construction is almost negligible. Users can choose video segmentation models of different sizes based on the latency requirements of their applications. Thank you for your valuable suggestion, and we will include this discussion in the revised version.
>
> > [Q1] Discussion on the potential bias and hallucination introduced by GPT-4o.
>
> Thank you for raising this important concern which is indeed a common challenge faced by all works foucused synthetic data.
>
> To address this, we first conducted small-scale trial annotation (hundreds of samples), followed by manual review to assess annotation quality and identify common hallucination or bias patterns. Based on these observations, we refined the task prompts and optimized the annotation pipeline. Large-scale annotation was only launched once we were satisfied with the output quality.
>
> During initial trials, we observed two main issues: (1) When scenes contained many instance objects, the model occasionally confused their attributes or relationships; (2) For longer videos, it exhibited temporal hallucinations and struggled to maintain accurate spatiotemporal grounding of specific objects.
>
> To mitigate these issues, we introduced several strategies:
> * Rather than directly annotating raw videos or images, we inserted SoM visual prompts and used instance IDs to guide the model’s attention. This significantly improved GPT-4o’s spatial and temporal grounding, even in complex scenes.
> * For long video sequences, we adopted a sliding-window approach with adjacent frames and explicitly prompted the model to focus on local temporal dynamics, which effectively reduced long-range hallucinations.
> * During training, we carefully balanced general data with INST-IT to prevent overfitting to a narrow distribution. Furthermore, we froze the first 12 layers of the vision encoder to avoid biasing the model toward visually prompted inputs.
>
> These efforts substantially alleviated the hallucination risk. Performance gains on both general benchmarks and instance-level understanding tasks further demonstrate the this claim.
>
>
> > [Q3] Typical duration of videos and frame sampling strategy; risk of missing important frames.
>
> * **Video duration.**
> Specifically, the INST-IT dataset contains 21,000 video examples, with an average duration of 18.4 seconds and a maximum of 234 seconds. Among them, 5,580 videos are longer than 30 seconds, and 1,483 videos are longer than 60 seconds.
> We would like to respectly note that the primary goal of INST-IT is to support instance-level understanding, which prioritizes the richness and precision of fine-grained annotations over the length of the videos.
>
> * **Frame sampling strategy.**
> For videos sourced from VidOR, keyframes are determined based on the original human annotations; for videos from other sources, we adopt a fixed-interval frame sampling strategy.
> Indeed, applying a uniform sampling interval across all videos may risk missing critical frames. To mitigate this, we manually inspected a random subset of videos from each source. Based on differences in visual information density and video characteristics, we then assigned different sampling intervals tailored to each source. We present the annotation intervals (annotated frames per second) for different sources in the table below. The average annotation frequency in INST-IT is **0.74 FPS**, which is significantly denser than that of previous works such as LLaVA-Hound (0.008 FPS) `[5]` and ShareGPT4Video (0.15 FPS) `[6]`. This higher density substantially reduces the likelihood of missing critical frames.
>
> |     | BenSMOT | LVVIS | OVIS | BRUST | UVO | VidOR | YoutubeVIS-2021 | Avg |
> | :---: | :-------: | :-----: | :----: | :----: | :---: | :----: | :----: | :----: |
> | **FPS** |  1/6   |  1  |  1/2 |  1/2  |  1 |  - |  4/5 | **0.74** |
>
> > [Q4] The total cost of annotating the entire dataset.
>
> We use `GPT-4o` API to provide detailed and dense annotations for 51K images and 21K videos, and the total API cost is approximately $6000.
>
> ---
>
> Thank you again for your valuable feedback and professional comments. We have done our best to address your concerns and hope our responses are satisfactory. Please feel free to reach out to us if you have any further questions. If you feel they are resolved, we would sincerely appreciate your re-evaluation of our work~ Looking forward to hearing from you!
>
> Best regards,
>
> The authors
>
> **References**
> 1. LMMs-Eval: Reality Check on the Evaluation of Large Multimodal Models
> 2. Set-of-Mark Prompting Unleashes Extraordinary Visual Grounding in GPT-4V, Arxiv 2023
> 3. ViP-LLaVA: Making Large Multimodal Models Understand Arbitrary Visual Prompts, CVPR 2024
> 4. SAM 2: Segment Anything in Images and Videos, Arxiv 2024
> 5. Direct Preference Optimization of Video Large Multimodal Models from Language Model Reward, Arxiv 2024
> 6. ShareGPT4Video: Improving Video Understanding and Generation with Better Captions, NeurIPS 2024

---

> > ### Comment · Reviewer_no3B · 2025-08-05
> >
> > Thank you for the detailed responses. I hope these clarifications and improvements will be incorporated into the revised version.

---

> > > ### Author Response · Authors · 2025-08-05
> > > **Thank you very much for your reply!**
> > >
> > > Dear reviewer,
> > >
> > > Thank you very much for your reply. We promise to incorporate the clarifications and improvements into the revised version. Please let us know if you have any further questions to discuss. **If our response addresses your concerns, we’d sincerely appreciate your consideration in raising the score. Thank you very much :- ) !**
> > >
> > > Best regards,
> > >
> > > The authors

---

### Official Review · Reviewer_42ms · 2025-07-02

**Clarity:** 2
**Significance:** 3
**Originality:** 2
**Rating:** 4
**Confidence:** 5

**Summary:**

This paper addresses a significant limitation in current MLMMs: their difficulty with "instance-level understanding," which requires comprehending the specific details, attributes, and relationships of individual objects within images and videos, as opposed to understanding the scene holistically. To tackle this, the authors introduce INST-IT, a comprehensive solution comprising a dataset, a benchmark, and the trained models. Experiments show that models enhanced with INST-IT achieve state-of-the-art performance on the INST-IT Bench and other instance-understanding benchmarks like RefCOCOg and ViP-Bench.

**Questions:**

Generally, I think this paper is good, and please refer to the weaknesses part.

**Ethical Concerns:**

["NO or VERY MINOR ethics concerns only"]

**Final Justification:**

I think this work contributes a good instance-level dataset for MLLM. I will vote for accept this paper.

**Quality:**

2

**Strengths And Weaknesses:**

Strengths:
1. The instance-level understanding is a missing puzzle for existing MLLMs datasets and benchmarks.
2. The dataset/benchmark curation pipeline is technically sound.
3. The paper provides extensive experiments and ablation studies.

Weakness:

I think it's not that natural for users to ask questions by giving a lot of visual masks. Instead, I think it's more common to use language to specify different people or objects in the video. I can understand this might be harder to build benchmarks or datasets, but It would be great if the author could discuss this.

---

> ### Author Rebuttal · Authors · 2025-07-31
>
> Dear reviewer 42ms,
>
> We sincerely appreciate your valuable feedback and insightful comments, which are greatly helpful in improving our work. We are highly encouraged that you thought our paper is good. We will address your concerns as follows.
>
> ---
>
> > [W1] Regarding how to refer to objects in videos, visual prompts or language descriptions?
>
> Great question! For instance-level understanding, how to inform the model of the instances we are interested in is an important design consideration. The main approaches include 1) visual prompting and 2) language referring. Each method has its own advantages and limitations.
>
> * Using visual prompts allows for explicitly and accurately marking the region of interest in the frame, which can more directly guide the model's attention to the specified area. However, the labor required to annotate visual prompts is indeed an issue that needs to be considered.
> * Using language to refer to specific instances is more natural and aligns better with human communication habits. However, due to the potential ambiguity of language, such references might become unclear when there are many objects. This also places higher demands on the model’s grounding ability to locate the region referred to by language within the visual scene.
>
> Additionally, in terms of user experience, with the rapid advancement of interactive segmentation paradigms such as Segmentation Anything (SAM `[1,2]`) and SEEM `[3]`, users can now directly generate object masks by simplely clicking or swiping, offering a natural and intuitive interaction to work with visual prompts.
>
> It is worth noting that although INST-IT is trained using explicit visual prompts, our model also exhibit improved understanding of language-based references. For instance, in the example shown on the left side of Figure 1 in the main paper, we directly use terms such as “man” and “woman” for instance reference, and our model demonstrates better comprehension compared to other models. We speculate that this improvement may stem from the grounded instance-level annotations in INST-IT, which implicitly enhance the model’s ability to interpret referring expressions and perform visual grounding.
>
> Thank you for bringing this up. We will include a discussion on this point in the revised version.
>
> ---
>
> We sincerely thank you once again for your insightful questions and are glad to have a meaningful discussion with you. We have shared our thoughts on instance referring in videos, and hope to engage in further discussion with you:-) Looking forward to your reply!
>
>
> Best regards,
>
> The authors
>
> **References**
> 1. Segment Anything
> 2. SAM 2: Segment Anything in Images and Videos
> 3. Segment Everything Everywhere All at Once

---

> > ### Comment · Reviewer_42ms · 2025-08-04
> >
> > Thanks for the detailed explanation from the authors. I don't have any further questions. I will keep my positive score.

---

> > > ### Author Response · Authors · 2025-08-08
> > >
> > > Dear Reviewer,
> > >
> > > Thank you very much for your response. We are very glad that you have no further questions, and we will certainly incorporate the corresponding content in the revised version. We sincerely appreciate your positive assessment, which encourages us greatly. We would be very grateful if you might consider supporting the acceptance of our paper and, if appropriate, raising your rating : -) !
> > >
> > > Best regards,
> > >
> > > The authors

---

### Official Review · Reviewer_SMig · 2025-07-02

**Clarity:** 3
**Significance:** 2
**Originality:** 2
**Rating:** 3
**Confidence:** 4

**Summary:**

This paper addresses the limitation of Large Multimodal Models (LMMs) in instance-level understanding, which requires fine-grained comprehension of specific objects within images and videos. The authors propose INST-IT, a comprehensive visual instruction-tuning dataset annotated by GPT-4o with Set-of-Marks (SoM) visual prompts, resulting a large-scale instruction-tuning dataset with 51k images and 21k videos. The work also introduces INST-IT Bench for evaluating instance-level understanding capabilities. Experiments show that training on INST-IT can improve the performance on image understanding and video understanding QA, proving the quality of the dataset.

**Questions:**

1. How much effects does SoM have on the data quality? Are there any experiments proving it?
2. For videos, how does the method ensure temporal consistency of instance IDs across frames, especially when instances appear and disappear?

**Ethical Concerns:**

["NO or VERY MINOR ethics concerns only"]

**Final Justification:**

Thanks for the rebuttal, it solves some of my concerns.

**Limitations:**

yes

**Quality:**

3

**Strengths And Weaknesses:**

## Strengths
1. This paper focus on the speicfic instance understanding task and curated instruction-tuning dataset to improve it. Understand specific instances rather than just holistic scene understanding is practically important for real-world applications.
2. The use of GPT-4o with SoM visual prompts to generate multi-level annotations (instance-level, frame-level, temporal dynamics, video-level) is creative and appears effective.
3. Thorough evaluation on both image understanding benchmarks and video QA benchmarks, with consistent improved performance after adding INST-IT as the benchmark.

## Weaknesses
1. The paper's method novelty is dataset itself, with little analysis of potential VLM architecture exploration (directly use LLaVA-Next).
2. Also the new dataset is also curated by GPT-4o, thus making the whole method like a distillation from GPT-4o. It's unknown whether the SoM can improve the data curation quality, or the whole performance improvement simply benefit from the GPT-4o's ability.

---

> ### Author Rebuttal · Authors · 2025-07-31
>
> Dear reviewer SMig,
>
> Thank you for your time and effort, which are greatly appreciated in helping us improve our work. We will address your concerns one by one.
>
> ---
>
> > [W1] Method novelty
>
> While the dataset is indeed one of our key contributions, our work goes beyond data collection in several important aspects. Specifically, we
> (1) investigate the effectiveness of instruction tuning with explicit visual prompts for enhancing fine-grained spatiotemporal understanding; and
> (2) propose a continuous instruction tuning paradigm that includes carefully designed data combination strategies to balance general capabilities and instance-level understanding, as well as a tuning scheme tailored for the visual encoder.
>
> Rather than modifying the model architecture itself, our research focuses more on developing a model-agnostic training framework aimed at enhancing multimodal instance understanding.
>
>
> > [W2, Q1] The effectiveness of SoM in improving data quality
>
> Thank you for raising this question. In fact, SoMs visual prompts play a critial role in instance-level annotation by marking numerical IDs on each instance in the frames and use the corresponding ID in to reference them. This significantly enhances GPT-4o’s grounding ability, enabling it accurately focus on the specific instance of interest.
>
> Without SoMs, GPT-4o struggles to produce fine-grained annotations that meet the requirements, the key challenges are:
> * GPT-4o struggles to precisely locate target instances, especially in crowded scenes or when occlusion occurs.
> * In video frames lacking instance ID prompts, GPT-4o struglle to recognize an object as the same one after it disappears and reappears.
>
> By providing a straightforward referencing (direct use of instance IDs) and explicit spatiotemporal grounding, SoMs effectively enable fine-grained spatiotemporal annotation at the instance level, thereby improving annotation quality.
>
> To further investigate, we replace the SoMs in the INST-IT Dataset with language descriptions, to simulate model training without visual prompts. As shown in the table below, the model trained with SoM annotations outperforms the one without SoMs, confirming its effectiveness.
>
> |                                          | MMMU | POPE | RefCOCOg(CIDEr) | INST-IT-I |
> |----------------------------------------------|------|------|-----------|--------------------|
> |  /wo SoM         | 36.7 | 87.0 | 60.5      | 49.5               |
> | /w SoM           | 37.8 | 87.5 | 62.0      | 59.6               |
>
>
> > [Q2] How to ensure the temporal consistency of instance IDs across frames
>
> Maintaining the consistent instance IDs across frames is indeed a crucial concern.
> To ensure high data quality, we use video segmentation and multi-object tracking datasets, in which the IDs are **manually annotated**.
> This guarantees the temporal consistency, even when instances disappear or reappear.
>
> In addition, recent advances in video instance segmentation `[1][2][3]` have shown strong performance in ID matching across frames, especially when instances disapper and reappear. These models can serve as promising tools to annotate instance IDs automatically, and we leave this exploration for future work.
>
> ---
>
> Thank you once again for your valuable feedback and thoughtful comments. We hope that our responses can address your concerns. If you have any additional questions, please feel free to reach out to us. If you feel your concerns are resolved, we kindly hope that you can consider reevaluating our work. We look forward to your reply!
>
> Best regards,
>
> The authors
>
> **References**
> 1. Matching Anything by Segmenting Anything, CVPR2024
> 2. SAMURAI: Adapting Segment Anything Model for Zero-Shot Visual Tracking with Motion-Aware Memory, Arxiv 2024
> 3. SeC: Advancing Complex Video Object Segmentation via Progressive Concept Construction, Arxiv 2025

---

### Official Review · Reviewer_67yH · 2025-07-02

**Clarity:** 4
**Significance:** 4
**Originality:** 3
**Rating:** 4
**Confidence:** 4

**Summary:**

This paper proposes INST-IT, a framework for instance-level understanding via explicit visual prompt Instruction Tuning. It introduces both the INST-IT benchmark and a large-scale training dataset. Experiments demonstrate that training on this dataset—combined with LLaVA-NeXT-Data—significantly improves performance on the INST-IT benchmark, as well as general video and image understanding benchmarks. These improvements are observed when following the authors' proposed training recipe, data combination strategy, and layer tuning scheme.

**Questions:**

Question.

1. Regarding Video-MME, could the authors provide a breakdown of performance across different axes such as *Overall*, *Medium*, and *Long* video segments? This would help clarify whether the proposed INST-IT dataset improves long video understanding.
2. For the INST-IT training dataset, could the authors report statistics on the duration of the video examples included? This would help better understand the dataset composition.
3. An open-ended question: many examples in the benchmark ask about the *exact ordering of frames*, such as “Who disappears from the frame at `<idx>`?”. However, many models apply temporal compression or frame subsampling before input, which makes precise frame index questions potentially unfair. Moreover, this may not be an optimal way to construct training data. Do the authors think it would be better to use **timestamps** or **relative temporal markers** (e.g., "beginning", "middle", "end") instead of raw frame indices in both training and evaluation?

**Ethical Concerns:**

["NO or VERY MINOR ethics concerns only"]

**Final Justification:**

The authors' rebuttal addressed my questions. The paper’s primary contribution lies in the dataset itself, with limited technical novelty and contribution in the method or model architecture. Therefore, I will maintain a borderline acceptance.

**Limitations:**

Yes

**Quality:**

4

**Strengths And Weaknesses:**

**Strengths**

- The paper is well-written and easy to follow.
- The proposed INST-IT training dataset and benchmark represent a substantial contribution to the community.
- Experimental results offer valuable insights—for instance, that high-quality instance-level understanding data can also enhance generic image and video comprehension.
- The dataset description is thorough, detailing a multi-level, instance-centric annotation pipeline and quality filtering procedures.
- The experimental evaluation is comprehensive, and the ablation studies provide useful guidance on balancing image and video data.

**Weaknesses**

While the paper is comprehensive and shows improved results, it seems more appropriate for the *Datasets and Benchmarks* track rather than the main NeurIPS track. The primary contribution is a new dataset, with relatively limited methodological innovation. Although the dataset appears highly useful, the work may fall short of the level of technical novelty typically expected for the main conference track.

---

> ### Author Rebuttal · Authors · 2025-07-31
>
> Dear reviewer 67yH,
>
> We sincerely appreciate your valuable feedback and insightful comments, which are greatly helpful in improving our work. We will address your concerns as follows.
>
> ---
> > [W1] Track appropriateness and primary contribution
>
> We appreciate the reviewer's feedback and are highly encouraged by the recognition of our dataset as highly useful and our paper is comprehensive.
>
> While the dataset is indeed an important part of INST-IT, we would like to clarify that our contributions go beyond data creation.
> Specifically, we
>
> * **Demonstrate the effectiveness of explicit visual-prompted instruction tuning**, showing that directly training with images and videos with visual prompts can effectively enhance the model’s fine-grained spatiotemporal understanding capabilities.
> * **Desgin a training recipe that enhances the model’s instance-level understanding without compromising its general capabilities**, which includes a detailed data combination strategy and an investigation of tunable layers within the vision encoder.
> * **Emphasize the importance of instance-level understanding**, a challenge that has been relatively underexplored compared to holistic scene-level understanding. Our work identifies key limitations in existing models and proposes INST-IT as a solution to this challenge.
>
> Therefore, our work not only introduces a novel dataset but also highlights the importance of instance-level understanding and proposes the use of explicit visual-prompted instruction tuning along with an effective training recipe to address this challenge.
>
> > [Q1] Breakdown of Video-MME performanceg
>
> Thank you for you suggestion. We present the performance breakdown across different axes of Video-MME in the table below.
>
> | Model | Short | Medium | Long | Overall |
> |-------|-------|--------|------|---------|
> | Baseline| 41.9 | 35.8 | **30.8** | 36.2 |
> | LLaVA-Next-Inst-It-Vicuna-7B | 53.1 (+11.2) | 42.2 (+6.4) | **38.0 (+7.2)** | 44.3 (+8.1) |
> | LLaVA-Next-Inst-It-Qwen2-7B | 63.0 (+21.1) | 51.3 (+15.5) | **45.7 (+14.9)** | 54.0 (+17.8) |
>
> The results demonstrate that the INST-IT Dataset effectively enhances the performance on VideoMME,
> including the long video understanding capabilities. When employing Vicuna-7B and Qwen2-7B as LLM backbones, we achieved improvements of 7.2% and 14.9% respectively on the Long segment of VideoMME.
>
> > [Q2] Statistics on the duration of the video examples
>
> Specifically, the INST-IT dataset contains 21,000 video examples, with an average duration of 18.4 seconds and a maximum of 234 seconds. Among them, 5,580 videos are longer than 30 seconds, and 1,483 videos are longer than 60 seconds.
>
> We would like to respectly note that the primary goal of INST-IT is to support instance-level understanding, which prioritizes the richness and precision of fine-grained annotations over the length of the videos. Moreover, since we use a sliding window over adjacent frames for temporal dynamic annotation, the data pipeline of INST-IT is also well-suited to long videos, we leave this extention for future works. Thank you for your suggestion. We will add the statistics on video durations to the revised manuscript.
>
> > [Q3] Temporal referencing method in both training and evaluation
>
> Thank you for the great question. In fact, timestamps and frame indices can be converted between each other. We believe that using timestamps is a more suitable choice for long video understanding, especially when leveraging LLMs with extended context lengths (e.g., 128k in Seed1.5-VL`[1]`), while frame indices are more appropriate for short video understanding and LLMs with relatively shorter context lengths (e.g., 16k/32k in LLaVA-NeXT`[2]`).
> Since our current dataset and benchmark focus on relatively short videos, we adopt frame indices for reference. We plan to explore long video understanding in future work.
>
>
> ---
>
> Many thanks to Reviewer 67yH for the professional and valuable reviews. We have try our best to address each of your concerns, and hope you can feel satisfied with our responses~ Please let us know if you have any further questions. Looking forward to hearing from you!
>
> Best regards,
>
> The authors
>
> **References**
> 1. Seed1.5-VL Technical Report, Arxiv 2025
> 2. LLaVA-NeXT: Improved reasoning, OCR, and world knowledge

---

> > ### Comment · Reviewer_67yH · 2025-08-05
> >
> > Thank you for the detailed responses. They addressed all my questions, though I still maintain that the paper’s primary contribution lies in the dataset. That said, I do recognize the value of this work and would like to retain my current positive rating.

---

> > > ### Author Response · Authors · 2025-08-08
> > >
> > > Dear Reviewer,
> > >
> > > Thank you very much for your response. We are very glad that we have addressed all of your questions, and we will certainly incorporate the corresponding content in the revised version. We sincerely appreciate your positive assessment, which encourages us greatly. We would be grateful if you might consider supporting the acceptance of our paper and, if appropriate, raising your rating.
> > >
> > > Best regards,
> > >
> > > The authors

---

### Decision · Program_Chairs · 2025-09-17

**Decision:**

Accept (poster)

**Comment:**

The final ratings are mixed (two "Borderline Accept," two "Borderline Reject").
After a careful review of the paper, rebuttal, and discussions, the AC leans toward acceptance.

The AC agrees with reviewers that the paper addresses an under-explored yet important limitation in current VLLMs: fine-grained, instance-level understanding. The core contribution, the INST-IT dataset and benchmark, was consistently recognized as a high-quality and valuable resource.

However, initial reviews raised valid concerns. The most significant issues were:
1) Methodological Novelty: Several reviewers (67yH, SMig, no3B) noted that the primary contribution is a dataset and training recipe rather than a novel model or algorithm.
2) Data Curation Effectiveness: Reviewer (SMig) questioned whether performance gains were simply from distilling knowledge from GPT-4o, asking for more evidence on the specific contribution of the Set-of-Marks (SoM) visual prompts.

The authors' rebuttal effectively addressed these points. They justified their contribution by framing it as a complete, model-agnostic training paradigm and benchmark (Inst-IT), validated it with an additional baseline (Qwen2.5-VL-3B-Instruct). Also, they quantified the impact of SoM visual prompts with an ablation study, demonstrating their value beyond simply using GPT-4o.

While the concern regarding methodological novelty is valid, the AC believes that the paper already makes a valuable contribution by identifying a key problem, creating a high-quality benchmark to diagnose it in current state-of-the-art VLMs, and demonstrating a successful training paradigm that boosts performance across various models. The authors' rebuttal and additional experiments have addressed the paper's initial weaknesses. Therefore, the AC recommends this paper for acceptance.